# Nicotine and Cytisine Embryotoxicity in the Experimental Zebrafish Model

**DOI:** 10.3390/ijms241512094

**Published:** 2023-07-28

**Authors:** Wojciech Świątkowski, Barbara Budzyńska, Monika Maciąg, Agnieszka Świątkowska, Przemko Tylżanowski, Mansur Rahnama-Hezavah, Piotr Stachurski, Renata Chałas

**Affiliations:** 1Department of Oral Surgery, Medical University of Lublin, 20-059 Lublin, Poland; mansur.rahnama-hezavah@umlub.pl; 2Independent Laboratory of Behavioral Studies, Medical University of Lublin, 20-059 Lublin, Poland; barbara.budzynska@umlub.pl (B.B.); monika.maciag@umlub.pl (M.M.); 3Department of Jaw Orthopaedics, Medical University of Lublin, 20-059 Lublin, Poland; agnieszka.swiatkowska@umlub.pl; 4Laboratory of Molecular Genetics, Department of Biomedical Sciences, Medical University of Lublin, 20-059 Lublin, Poland; przemyslaw.tylzanowski@umlub.pl; 5Department of Development and Regeneration, Skeletal Biology and Engineering Research Center, KU Leuven, 3000 Leuven, Belgium; 6Department of Developmental Dentistry, Medical University of Lublin, 20-059 Lublin, Poland; piotr.stachurski@umlub.pl; 7Department of Oral Medicine, Medical University of Lublin, 20-059 Lublin, Poland; renata.chalas@umlub.pl

**Keywords:** tobacco, pregnancy, developement, zebrafish, toxicity attenuation

## Abstract

Tobacco smoking is one of the most serious health problems. Potentially lethal effects of nicotine for adults can occur with as little as 30 to 60 mg, although severe symptoms can arise with lower doses. Furthermore, the route of administration also influences the toxicity. Cytisine is one of the most popular medications in nicotinism treatment. Like nicotine, cytisine is a plant alkaloid, signaling through nicotinic acetylcholine receptors. Our study evaluated the effects of cytisine in nicotine-induced embryotoxic effects using zebrafish larvae. We examined the teratogenicity of nicotine and cytisine alone or in combination. Nicotine increased mortality and delayed hatching of zebrafish larvae in a dose-dependent manner. Cytisine did not affect mortality in a wide range of concentrations, and hatching delay was observed only at the highest concentrations, above 2 mM. Administering compounds together partially reduced the adverse teratogenic effect induced by nicotine alone. The protective effect of cytisine against the nicotine effect, observed in zebrafish, will contribute to future studies or treatments related to nicotine addiction or prenatal nicotine exposure in humans.

## 1. Introduction

Tobacco use poses a significant threat to public health. WHO reports that each year, tobacco use claims the lives of over eight million individuals. In 2010, the estimated prevalence of smoking among women in high-income countries was approximately 21.2%, while in low-income countries, 8%. Projections indicate that by 2025, the prevalence of tobacco use among women in high-income countries is likely to decrease to around 17%, while in low-income countries, it is expected to reach a low estimate of 3.8% [1]. Data from the Pregnancy Risk Assessment Monitoring System (PRAMS) in the USA revealed that nearly a quarter of women of reproductive age were smokers before pregnancy, while roughly 10% continued smoking during the last trimester. It was also found that 55% of women who were smokers prior to pregnancy reported successfully quitting by the last trimester [2]. Kvale’s research has uncovered a correlation between smoking during pregnancy and several demographic factors, including young maternal age, unmarried status, or low socioeconomic status. The study found that mothers who fell within these categories were more likely to smoke during pregnancy, putting their unborn children at risk [3]. Studies have confirmed that maternal and/or passive smoking exposure during pregnancy increases the risk of wheezing and asthma in offspring as well as neurodevelopmental problems [2]. 

In infants, nicotine exposure raises the risk of Sudden Unexpected Infant Death Syndrome (SUID). In the USA, it is estimated that 22% of SUID’s can be linked to maternal smoking during pregnancy [4]. Nicotine has been shown to disturb brain development in prenatal, early postnatal, and adolescence in experimental animal studies [5]. Furthermore, smoking during pregnancy has been associated with various pregnancy complications and adverse outcomes in infants, including low birth weight. Clinical studies have identified a link between maternal smoking in early pregnancy and the development of specific birth defects in newborns, such as clubfoot, gastroschisis, atrial septal defects, craniosynostosis, and facial defects, as well as the occurrence of cardiovascular, gastrointestinal, and urogenital defects in children [6,7]. Nonetheless, there are very few studies on the exact impact of nicotine on embryogenesis and craniofacial development, for instance, the relationship between maternal smoking and orofacial malformations. 

Nicotine addiction treatment is most successful when it is tailored to the individual. Treatment options may include medication, counseling, and lifestyle changes. Cytisine can help reduce nicotine cravings, withdrawal symptoms, and the urge to smoke. This drug is a partial agonist of the α4β2 nicotinic acetylcholine receptor responsible for the central effects of nicotine. Unfortunately, the information regarding the use of cytisine in pregnancy or lactation is lacking. It has been shown that nicotinic receptors exist in the human prenatal brain already by the 4–5th week of gestation, which means that cytisine use during pregnancy could exert effects on the development of the nervous system. Embryonic development requires very precise spatiotemporal regulation of cellular proliferation and subsequent differentiation. During each of these stages, the embryo exhibits sensitivity to teratogens, which may result in lethality, whereas exposure during later stages is more likely to cause tissue malformations and/or neurological defects. Therefore, the focus on nicotine, one of many teratogenic factors, is so important [2].

The zebrafish (*Danio rerio*) is one of the most popular experimental models to study various human conditions, including addiction [8]. Zebrafish eggs are fertilized and develop ex utero, permitting studies of early development. Additionally, these fish grow at a rapid rate, developing as much in a day as a human embryo develops in one month, permitting, among others, neurological studies. Moreover, zebrafish have a similar genetic structure to humans. They share 70% of genes and most of the metabolic pathways with humans [9].

Since studies on pregnant women are not typically carried out, we used the zebrafish model as an acknowledged proxy to evaluate the safety of cytisine and nicotine using zebrafish larvae.

## 2. Results

### 2.1. Nicotine-Induced Toxicity

First, we wanted to evaluate the nicotine toxicity in our model. Therefore, we evaluated the effect of nicotine exposure using the Fish Embryo Acute Toxicity (FET) Test [10]. At 96 hpf, nicotine had low toxicity, resulting in LC_50_ of 0.61 mM (Figure 1A). The delay of hatching rate was observed at 72 h post-fertilization (hpf) at the concentrations of 0.5, 0.75, and 1.0 mM in 8%, 12.5%, and 28.6% of fish, respectively (Figure 1B). To explore specific toxic effects of tested compounds on the cardiovascular and central nervous systems, as well as morphological alterations, we analyzed changes in heart rate, locomotor activity, and scored types of developmental malformations. Nicotine had no effect on heart rhythm (Figure 1C) nor distance or larvae velocity swim (Figure 1D,E). It did, however, delay the development of pigment (PIG) at the concentrations of 0.5, 0.75, and 1.0 mM in 27.3%, 25%, and 71.4% of animals, respectively (Figure 1F,G). For further studies, the sublethal or lethal doses that impaired the hatching rate, and caused morphological alterations (i.e., 0.5, 0.75, and 1.0 mM) were selected.

### 2.2. Cytisine Toxicity in the Zebrafish Embryo

Next, we wanted to compare cytisine and nicotine toxicity. Cytisine did not induce mortality in a wide range of concentrations (0.01–8 mM) for up to 96 hpf (Figure 2A). The delay of the hatching was observed at 2.0 mM in 67% of animals, reaching 100% at doses above 4.0 mM (Figure 2B). Only at the highest tested dose of 8.0 mM, the heart rate was decreased (Figure 2C). Cytisine reduced the development of pigment at concentrations starting from 2.0 mM (Figure 2D). Based on the lack of mortality, no changes in hatching rate, nor morphological development, the dose of 1.0 mM was selected for further studies.

### 2.3. Cytisine Ameliorates Nicotine-Induced Toxicity

In a typical patient situation, during the period of quitting smoking, the patient is exposed to both nicotine and cytisine. Therefore, we wanted to explore the combined effect of both alkaloids on zebrafish development. As expected, nicotine alone elicited mortality in a dose-dependent manner at doses 0.75 and 1.0 mM (*p* < 0.01, 0.001, respectively), whereas cytisine alone (1.0 mM) did not induce mortality (Figure 3A). Co-treatment of zebrafish with cytisine and nicotine at a dose of 0.5 mM synergistically and significantly increased the mortality rate from 8.3% to 25.0%. The nicotine-induced mortality at the doses of 0.75 and 1.0 mM was attenuated by cytisine by 29.2% (ns vs. nicotine 0.75 mM) and 83.3% (*p* < 0.01 vs. nicotine 1.0 mM), respectively (Figure 3A). Furthermore, microscopic examination showed that zebrafish larvae exposed to cytisine and 0.5 mM of nicotine had signs of toxicity, including pericardial edema (PE), spinal curvature (SC), and reduction of pigment (PIG), while the co-treatment with 1.0 mM of cytisine protected against all malformations except pigment depletion (Figure 3A,B).

Administration of nicotine significantly reduced heart rate (0.75 mM, *p* < 0.05, 1.0 mM *p* < 0.001 (Figure 3C), and cytisine co-exposure statistically significantly increased the observed parameter (*p* < 0.01 nicotine 1.0 mM vs. cytisine/nicotine group). Similar effects were observed regarding distance and velocity. Nicotine at the concentration 1 mM decreased both parameters (*p* < 0.05). Zebrafish co-treated with Cytisine reversed nicotine signs of neurotoxicity assessed by a locomotor response (*p* < 0.001) (Figure 3D,E). 

## 3. Discussion

We investigated the toxic impact of cytisine on nicotine-induced embryotoxicity using a zebrafish model. A comparison of these two alkaloids showed that cytisine is less teratogenic than nicotine, at all parameters tested. The negative effects, such as a decrease in hatching or reduced development of pigment, were observed only at higher concentrations of cytisine, starting from 2.0 mM. For nicotine, these changes were observed at the lower concentrations starting from 0.5 mM. Co-exposure to toxic concentrations of nicotine (0.5–1.0 mM) with a nontoxic concentration of cytisine (1.0 mM) decreased the occurrence and intensity of teratogenic effects induced by nicotine. Mortality and heart rate of the embryos in the nicotine group were reduced by co-incubation with cytisine. We observed decreased mortality and toxicity defined as pericardial edema, spinal curvature, and disturbances in pigment in groups treated with nicotine and cytisine in comparison with nicotine-treated zebrafish. Although altered by nicotine (0.75 mM), heart rate was not changed by cytisine.

The attenuating effect of cytisine on nicotine-induced teratogenicity is novel and intriguing. It opens a possibility to further explore it preclinically in rodents to observe the influence of cytisine on the teratogenic effect of nicotine on the fetus.

There are variations in the biological half-life of nicotine in different populations. Specifically, the biological half-life of nicotine in pregnant or postpartum women has been found to be 1.6 h and 1.8 h, respectively. This is faster than the biological half-life of nicotine in non-pregnant humans, which is around 2 h. Importantly, the biological half-life of nicotine in newborns three to four times longer, as the metabolism of nicotine via CYP2A6 is not well developed [11,12].

Chronic nicotine intoxication during pregnancy especially affects the nervous system of the fetus, not only in actively smoking women but also in women exposed to “second-hand” smoke. Nicotine easily crosses the blood-placenta barrier and reaches a high level in fetal serum [13]. Children whose mothers smoke or are exposed to tobacco smoke show behavioral abnormalities, including memory and learning deterioration, attentional deficits, and lowered IQ [14,15]. Many human and animal studies revealed that nicotine neurotoxic effects observed in the fetus’s brain last through adolescence [13,16,17]. This alkaloid penetrates the brain of the fetus and influences nAChRs at the gene level [18]. Persistent overstimulation of cholinergic receptors induced neurodevelopmental changes. Interestingly, neurotoxic effects of prenatal nicotine exposure in animal models were ameliorated by pretreatment with a nAChRs α7 antagonist methyllycaconitine, but not with a nAChRs α4β2 antagonist dihydro-β-erythroidine [19]. Thus, it was concluded that mainly homomeric α7 nAChRs are involved in the central effects of nicotine in fetuses. Furthermore, prenatal nicotine exposure induced acetylcholine transferase deficits, an enzyme responsible for the synthesis of ACh [20]. 

Cytisine is a partial agonist of the α4β2 subtype and a stronger agonist of the α3β4 and α7 subtypes. By activation of the nAChRs in the mesolimbic system (also known as the reward system), it inhibits the sensory stimulation of nicotine and decreases withdrawal symptoms [21]. It was revealed that LD_50_ for oral administration in rats is between 5 and 50 mg/kg. In 1912, the first reports concerning cytisine embryotoxic or teratogenic activity in animal studies showed that nicotine and cytisine compete and have different potencies in different organs, even though nicotinic receptors were described almost 110 years later [21,22]. Toxic effects observed in humans, particularly children, affect the gastrointestinal, central nervous, and motor systems [23]. Aside from toxicological effects related to nicotine, cytisine exhibits a moderate level of activity as an acetylcholinesterase (AChE) inhibitor [24]. As cytisine is an effective drug in the treatment of nicotinism and, at the same time, safe in therapeutic doses, the question arises as to whether it can be used in the treatment of nicotinism in pregnant women. Such studies have not been carried out so far, and the *Danio rerio* model is the most convenient model for the evaluation of the embryotoxicity of the tested compounds. 

Cholinergic cells are scattered in the zebrafish brain in a manner observed in other vertebrates, e.g., in the olfactory bulb, the autonomic ganglia, the motor nuclei of the skull, the cerebellum, the spinal cord, and the cerebellum [25]. So far, the existence of nine nAChR subunits has been demonstrated in zebrafish: α2, α3, α4, α6, α7, α8, β2, β3, and β4 [26]. At the same time, numerous subunits are observed several hours after fertilization. This knowledge can be useful, particularly in trying to estimate fish model utility as a model of human disease.

Our studies confirmed that nicotine is toxic with LC_50_ equal to 0.61 mM, for cytisine toxicity has not been established even at the maximal used concentration, 8 mM. We demonstrated that cytisine strongly inhibits hatching at doses that do not induce mortality. This may be related to the toxic effect on the neuromuscular system at the primary stages of zebrafish growth. It has been discovered that high doses of cytisine can cause muscle weakness and possible respiratory muscle paralysis as a result of inactivation of AChR in the muscles. Similar effects may be observed in zebrafish embryo muscle tone [27]. In the case of nicotine, the decrease in hatching occurs only at lethal doses.

Heart rate is one of the parameters defining cardiotoxicity. In our studies, nicotine had no effect, while cytisine showed potential cardiotoxic properties by decreasing the heart rate at the concentration of 8 mM. Previous research has demonstrated that cytisine triggers both parasympathetic and sympathetic responses in heart rate. It was reported in mice that intravenous administration of cytisine caused a significant drop in heart rate, which was then followed by subsequent rate increase [28]. This data indicates that the activation of α7 nAChRs is involved in the sympathetic pressor response. On the other hand, the activation of α4β2 nAChRs is crucial for the parasympathetic negative chronotropic effect, resulting in a decrease in heart rate. In our studies, we did not notice an increase in this parameter, but further studies are needed.

Thus, cytisine-induced bradycardia noticed in zebrafish larvae may result from the baroreflex-associated bradycardic response. We suggest that nicotine and cytisine co-administration induce a decrease in the heart rate by the aforementioned mechanism. Moreover, we did not observe changes in locomotor activity induced by nicotine, although other authors described such an effect [29]. In zebrafish, acute doses of nicotine may exert the opposite effect to chronic intoxication. For example, acute administration has been found to have anxiolytic properties, whereas chronic exposure elicits anxiogenic effects [30]. 

The effect of co-administration of nicotine and cytisine can be explained by their impact on nAChR. The available literature does not provide comprehensive information regarding the acetylcholine receptor (nAChRs) in zebrafish. However, it is known that zebrafish express α4, α2, and β2 subunits, indicating the likely presence of both α2β2 and α4β2 nAChRs. In terms of sensitivity, it has been observed that the efficacy and potency of nicotine and cytisine for specific nAChRs differ between zebrafish and mammals. Consistent with studies on mammalian nAChRs, nicotine has a relatively low potency and efficacy on muscle-type receptors. It exhibits high efficacy but low potency for α7 receptors while showing the best potency and good efficacy for α4β2 receptors. At the same time, cytisine is a full agonist for mammalian α7 and α3β4 receptors, but interestingly, it is only a full agonist for fish α7 receptors, displaying surprisingly low efficacy for α3β4 receptors [26]. In relation to the observed effect, it is known that zebrafish express α4β2 nAChRs, and the efficacy of cytisine for α4β2 receptors appears to be somewhat higher than what is typically reported for mammalian α4β2 receptors [26]. Therefore, it is tempting to speculate that the lack of a dose-dependent effect may be associated with the high potency of cytisine for this specific receptor. The toxic effect of the nicotine (0.5 mM) and cytisine (1.0 mM) group may result from a stronger cytisine affinity to α4β2 nAChR, which is heteromeric and can be responsible for different efficacy depending on the concentration of both alkaloids, which requires further studies. This phenomenon can be explained by the lower lipophilicity of cytisine. The most common method to treat fish larvae is to immerse them in the tested solution. Thus, the lipophilicity, as well as the molecular weight of the compound, might determine its activity. In our study, the lack of dose response might be therefore ascribed to increase absorption of cytisine into fish, binding to the receptors and lessening nicotine-induced toxic effects. It was also revealed that in rats’ brains, lower concentrations of cytisine were observed than those of equal doses of nicotine. Cytisine, when administered orally or intravenously, does not reach the brain in amounts exceeding 30% of its plasma concentration. In contrast, under the same conditions, nicotine reaches a brain concentration equivalent to 65% of its plasma concentration. These results suggest that cytisine has a limited ability to cross the blood-brain barrier (BBB) [31].

Significantly reduced mortality in groups incubated in the concentrations of 0.75 mM of nicotine together with cytisine (1.0 mM) may suggest the protective role of cytisine in nicotine-induced ecotoxicity. 

Our study on the zebrafish model also revealed a very interesting effect of cytisine and nicotine on pigmentation delay. In vertebrates, melanocytes are developed from neural crest cells. These cells differentiate among others, towards neurons of the peripheral nervous system, pigment cells, and craniofacial mesenchyme, forming cartilages of the facial skeleton [32].

Research has shown that exposure to nicotine can alter the pigmentation patterns in zebrafish. Nicotine acts as a stimulant and affects the functioning of melanocytes, the cells responsible for producing pigments [33]. The specific mechanisms by which nicotine influences pigmentation in zebrafish are still being investigated, but several studies have provided insights into this phenomenon. One study conducted on zebrafish embryos exposed to nicotine found that it led to a decrease in the number of melanocytes, resulting in reduced pigmentation. The nicotine exposure disrupted the migration of neural crest cells, which give rise to melanocytes during embryonic development. This disturbance in melanocyte development resulted in a visibly lighter pigmentation phenotype in the zebrafish [34]. Furthermore, nicotine exposure has been found to affect the expression of genes involved in pigmentation regulation. Certain genes, e.g., SLC24A5, that play a crucial role in melanocyte development and pigment production have shown altered expression patterns in response to nicotine exposure. These changes in gene expression contribute to the observed effects on pigmentation in zebrafish [35]. 

We observed disturbed pigmentation in all examined groups. This observation suggests that all neural crest-derived cells can be affected by both nicotine and cytisine. It seems to be one of the limitations of using cytisine in pregnant women due to many crucial organs and structures depending on neural crest embryogenesis. Therefore, we might suggest that it is worth examining nicotine and cytisine impact in this aspect in future studies.

Recent research has provided evidence that exposure to nicotine during embryonic development alters social behavior in adulthood and leads to craniofacial defects, resulting in reduced size of craniofacial cartilages. Similar metabolism of nicotine in zebrafish, as in humans, also confirms the very high application of this animal model [34]. The consequences of pigmentation alteration in mammals could be potentially very important. Due to the protective function of melanocytes in the skin, these effects also need to be evaluated.

## 4. Materials and Methods

### 4.1. Preparation of Drugs 

The drugs (−)-Nicotine hydrogen tartrate salt and cytisine were procured from Sigma-Aldrich (Sigma Aldrich, St. Louis, MO, USA) and their stock solutions were prepared in zebrafish medium with a pH of 7.1–7.3 and containing 5.0 mM NaCl, 0.17 mM KCl, 0.33 mM CaCl_2_, 0.33 mM MgSO_4_]. The final test concentrations (nicotine 0.1–1 mM, cytisine 0.01–8 mM) were achieved by diluting the stock solutions immediately prior to the experiments.

### 4.2. Zebrafish Maintenance

The zebrafish (*Danio rerio*) embryos were obtained from the animal facility located at the Experimental Medicine Center, Medical University of Lublin, Poland (the animal handling reference number is 03/OMD/OP/2022). Adult zebrafish of the AB strain were maintained under standard aquaculture conditions at 28.5 °C and a 14:10 h light/dark cycle. The zebrafish embryos were collected through natural mating.

### 4.3. Ethics Declarations

The experiments adhered to the guidelines set forth by the National Institute of Health for the Care and Use of Laboratory Animals and the European Community Council Directive for the Care and Use of Laboratory Animals of 22 September 2010. Additionally, all methods involving zebrafish embryos were in compliance with the Animal Research: Reporting of In Vivo Experiments (ARRIVE) guidelines. The experiment involving larvae up to 120 hpf did not require the agreement of the Local Ethical Committee. 

### 4.4. Chemical Treatment

The toxicity of chemical substances was evaluated using zebrafish embryos, based on OECD Guidelines for the Testing of Chemicals [10] with modifications according to Macig and co-authors [36,37]. To minimize delays, a minimum of 40 embryos were randomly selected for each treatment group and transferred to a 6-well plate containing 2 mL of either nicotine, cytisine, a combination of both drugs, or a selected concentration. Within 3 h post-fertilization, 96 viable fertilized embryos were individually transferred to 96-well plates (one fish per well) and incubated in 200 µL of either nicotine, cytisine, a combination of both drugs, or a control solution. The working solutions were refreshed every 24 h. The embryos were exposed to either the ‘control’ or ‘treatment’ solutions for up to 96 h. The plates were kept in an incubator at 28.5 ± 0.5 °C with a 14:10 h light/dark cycle. The embryos were observed under a light microscope (Stemi 508, Zeiss, Göttingen, Germany) before transferring them to the 96-well plates. After the experiment, the larvae were euthanized by immersion in a 15 µM tricaine methanesulfonate (Sigma Aldrich, St. Louis, MO, USA) solution. Although zebrafish are social animals, and their interactions with other fish can influence their behavior, and physiological response, we may suggest that separate rearing of zebrafish embryos did not affect the results of FET, as it was previously revealed [38]. To address these concerns and minimize bias, we implemented the following strategies: Randomization and appropriate control groups that are raised and treated identically to the experimental groups, except for exposure to the toxic agent, are crucial for establishing a baseline for comparison.

### 4.5. Toxicity Assessment

The survival, hatching rate, and developmental abnormalities of the embryos were recorded every 24 h under a stereomicroscope (Zeiss Stemi 508 microscope; Göttingen, Germany). At 96 hpf, the number of dead embryos was recorded, and the concentration that was lethal to 50% of the animals (LC_50_ value) was calculated. Embryos that had coagulated or showed no heartbeat for 20 s were considered dead and were discarded. A test was deemed valid only when the mortality in the negative control did not exceed 10%.

### 4.6. Morphological Analysis, Heart Rate and Locomotor Activity Assessment

Throughout the experiment, larvae were observed morphologically using a light microscope (Zeiss Stemi 508 microscope). At 96 hpf, the percentage of malformations, including pericardial edema, stretched heart, hemorrhage, head malformation, tail malformation, growth retardation, and/or pigment depletion, was determined as the ratio of abnormal zebrafish to the number of alive zebrafish. For imaging purposes, zebrafish larvae were immobilized in 3% methylcellulose, and photographs were taken using a Zeiss SteREO Discovery.V8 microscope. At 72 hpf, the ratio of unhatched and hatched fish was determined.

At 96 hpf, the heart rate of the larvae was measured. After being equilibrated at room temperature for 30 min, the heartbeats were counted for 20 s under a stereomicroscope. Within a 10-min light period at 96 hpf, the distance and mean velocity moved by the larvae were analyzed using EthoVision XT15 software (Version number: 15.0.1418; Noldus, Wageningen, The Netherlands).

### 4.7. Statistical Analysis

The dose-response mortality data were analyzed using nonlinear regression with a four-parameter sigmoidal curve to calculate the LC_50_ value. The statistical analysis was performed using one-way or two-way analysis of variance, followed by Tukey’s or Bonferroni’s post-hoc test, respectively, with a significance level of * *p* < 0.05. The results were presented as mean values. The analysis was conducted with Prism v8.3.1 software from GraphPad.

## 5. Conclusions

In conclusion, tobacco smoking poses a significant health problem, and nicotine toxicity can occur even at low doses, with potential lethality ranging from 30 to 60 mg for adults. The embryotoxic dose of nicotine in humans has not been determined, while the route of administration influences its toxicity. Many smokers, including pregnant women, are aware of the dangers associated with tobacco use but struggle to quit due to nicotine’s highly addictive nature. Our study focused on evaluating the effects of cytisine in nicotine-induced effects using zebrafish larvae as an animal model frequently utilized in investigating teratogenic effects. We found that nicotine increased mortality and caused delayed hatching in zebrafish larvae in a dose-dependent manner. On the other hand, cytisine did not significantly impact mortality across a wide range of concentrations, and only the highest concentrations caused a delay in hatching. Interestingly, when administered together, cytisine partially mitigated the adverse teratogenic effects induced by nicotine alone. The protective effect of cytisine observed in zebrafish larvae holds promise for informing future studies and potential treatments targeting nicotine addiction and prenatal nicotine exposure in humans. However, this finding could be further explored in more complex mammalian models to assess the potential of cytisine as a therapeutic agent for nicotine-induced embryotoxicity and safety in pregnancy. Further studies could focus on the molecular mechanisms by which cytisine reduces nicotine toxicity. Additionally, the prenatal safety of cytisine, as well as its potential side effects on the fetus, should be evaluated.

## Figures and Tables

**Figure 1 ijms-24-12094-f001:**
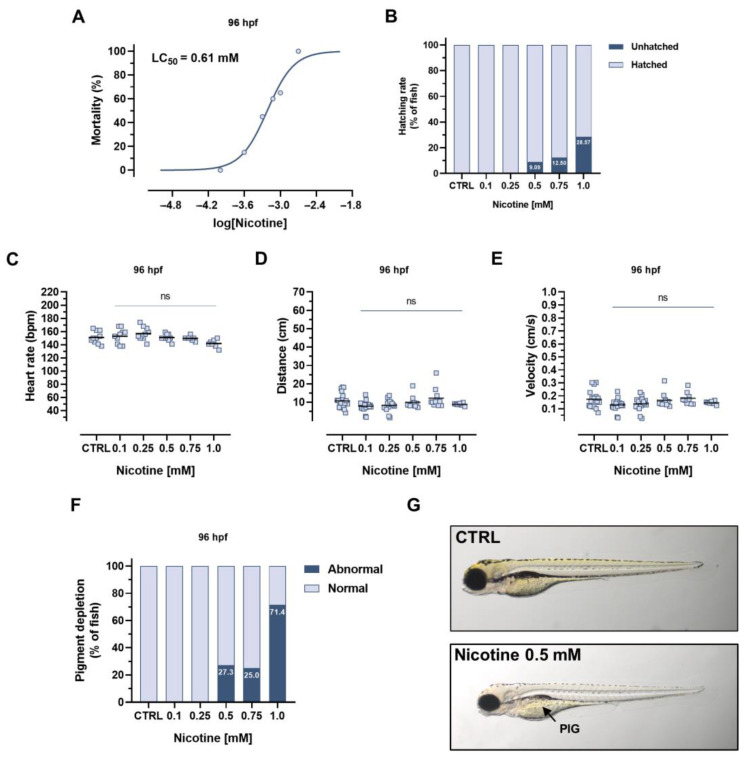
Toxicity assessment of zebrafish larvae exposed to nicotine. Effect of nicotine exposure on (**A**) mortality, (**B**) hatching, (**C**) heart rate, (**D**) distance, (**E**) velocity swam by larvae, and (**F**) morphological alterations, (**G**) representative images of 96 hpf zebrafish larvae co-treated with control and nicotine are provided. Nonlinear regression was performed on dose-response mortality data. Heart rate, distance, and velocity data are presented as mean and were assessed using one-way ANOVA. For mortality, hatching rate, and morphological changes data: *n* = 20, for heart rate and locomotor activity data: *n* = 7–10. Blue circle shows mean value for mortality data; blue square shows individual values, ns- not statistically significant values vs CTRL (control) group. PIG: Pigment Depletion.

**Figure 2 ijms-24-12094-f002:**
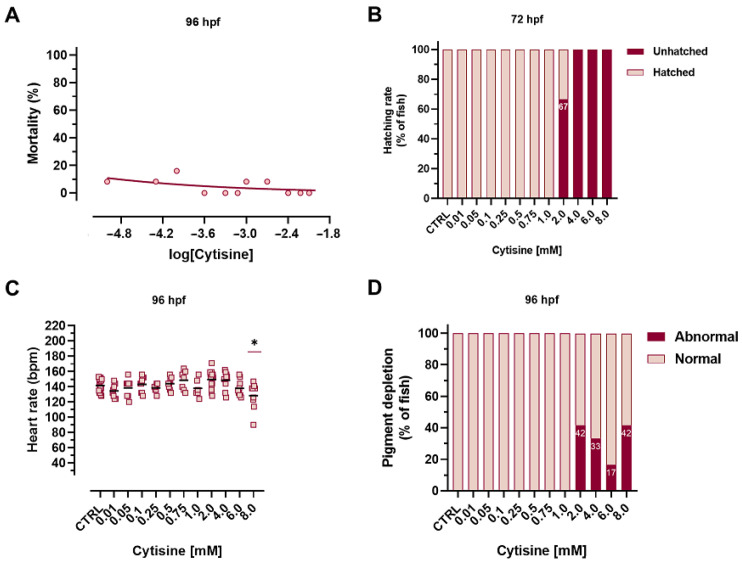
Toxicity assessment of zebrafish larvae exposed to cytisine. Effect of cytisine exposure on (**A**) mortality, (**B**) hatching rate, (**C**) heart rate, and (**D**) morphological alterations. Nonlinear regression was performed on dose-response mortality data (**A**). Hatching rate is presented as a ratio of unhatched and hatched fish scored at 72 hpf (**B**). Heart rate data are presented as mean and were assessed using one-way ANOVA followed by Tukey’s post-hoc test (**C**). The percentage of fish with pigment depletion was scored at 96 hpf (**D**). The confidence limit of * *p* < 0.05 was considered statistically significant. For mortality, hatching rate, and morphological changes data: *n* = 20, for heart rate data: *n* = 6–17. Red circle shows mean value for mortality data; red square shows individual values.

**Figure 3 ijms-24-12094-f003:**
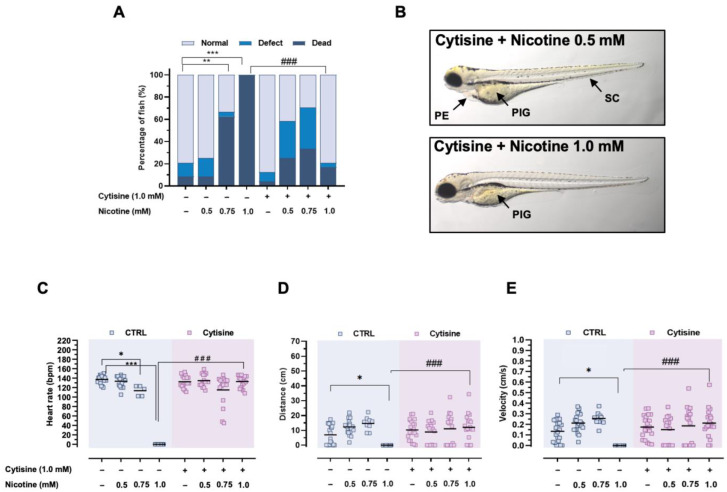
Protective assessment of cytisine against nicotine-induced toxicity. Effect of co-treatment of nicotine and (**A**) cytisine on mortality morphological alterations, (**B**) representative images of 96 hpf zebrafish larvae co-treated with cytisine and nicotine are provided, (**C**) heart rate, (**D**) distance, and (**E**) velocity swam by larvae. Heart rate, distance, and velocity data are presented as mean and were assessed using two-way ANOVA followed by Bonferroni’s post-hoc test. The confidence limit of * *p* < 0.05, ** *p* < 0.01, *** *p* < 0.001 vs. control group; ### *p* < 0.001 vs. nicotine treated group. For mortality and morphological changes data: *n* = 24, for heart rate and locomotor activity data: *n* = 5–20. PIG: Pigment Depletion. PE: Pericardial Edema. SC: Spinal Curvature. Squares show individual values.

## Data Availability

Not applicable.

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
