# Peer review of "Nicotine and Cytisine Embryotoxicity in the Experimental Zebrafish Model"

_ijms, 2023, doi:10.3390/ijms241512094_

Round 1

Reviewer 1 Report

This paper evaluates the safety of cytisine and nicotine using zebrafish larvae and had a nice discussion about the developing fetus.

1. Explain the phenotype in a more detailed way. For example, Fig1F shows nicotine affects PIG and some literature has shown their connection.

2. Fig2 C D Need horizontal coordinate description

3. Fig3A Line143-146 Why does co-treatment of nicotine and cystine increase the mortality rate from 8.3% to 25.0% at a dose of 0.5 mM?but decresed at the dose of 0.75mM and 1mM?

4. The nicotine-induced toxicity at 1mM looks inconsistently in Fig1C and Fig3C.

5. Fig 3 C-E includes statistics significance between control and cytisine group.

Overall the english looks good.

Reviewer 2 Report

The abstract would benefit from a sentence or two stating whether concentrations were relevant to current application in humans, as well as highlighting the wider context of the study beyond ZFE research. I would also advise replacing the keywords 'cytisine', 'nicotine', and ' embryotoxicity' with words not used in the title, as this will improve SEO and visibility of the research. The introduction is overall well formulated and lays out the improtance of reearch clearly. However, it could benefit from re-working. Some paragraphs are rather repetitive and reiterate the same point multiple times (e.g. lines 56-61, 65-68). The results are outlined clearly, the figures are clean and legible, and the titles for each section are informative. The discussion is detailed and contains suitable references as far as I am aware of the current literature. However, some statements are made that might overreach to scope of the study (i.e. lines 180-182). The materials and methods are detailed, but I am missing the animal handling reference number. Moreover, an adapted version of the FET test is applied, but no mention is made of the adapatations, other than in the methods. This should be made clear throughout the manuscript. The conclusion is very brief and would benefit from more detail and a wider outlook.

Line 24: Sentence seems to be missing iformation? Consider rewriting

Line 39: Does this mean that more women of reprod. age are smoking in developed countries than underdeveloped ones?

Line 49ff: Consider moving the information about what nicotine exposure does to development, before discussing the prevalence of smoking in mothers.

Line 67: Consider removing "reduce nicotine cravings, withdrawl symptoms, and the urge to smoke", as this was stated in the previous sentence.

Line 77: What do you mean with "at later stages, it typically leads factors to malformation"? Please rephrase to make the statement clearer.

Line 82: Start sentence with "The"

Line 84: "outside the mother's body" can be shortened to "ex utero"

Lines 88-89: Merge sentences

Line 99: This is an interpretation of the results and should be moved to the discussion

Line 123: Why was this range selected for cytisine? Could higher concentrations have head to mortality and thus LC values? Was there a solubility issue?

Line 137: Remove " ." from the end of the figure caption please

Line 151: Avoid the use of "interestingly" or other descriptors in the results. This should be saved for the discusssion

Lines 180ff: You later state the link between pigmentation and neural crest cells, so it seems an overreach of the scope of this paper to state thet cytisine could be used for pregant women if you clearly show it causing PIG in combination with Nicotine (figure 3). Consider moving this statement to the end of the discussion and refering to potential limitations, in order to avoid misrepresenting evidence.

Line 204: Consider replacing "Follow" with "According to" or something similar, as the grammar is currently incorrect.

Line 244: Please replace "coauthors" with "co-authors"

Line 303: Consider stating the exposure concentration. The method should be easy to follow without reading the rest of the article.

Line 310ff: Please include the animal handling license number somewhere within the paragraph for handling and keeping adult fish.

Line 321: You state you follow the FET test, but 6-well plates are not used for that. Please re-phrase the manuscript to highlight that you applied an adapted method. The number of individuals and replicates does not become clear to me. Are you transfering 20 invidials per 96-well plate well? Or are you rearing single embryos in wells? That significantly impacts development and behaviour! If this is indeed how they have been reared, I suggest running one replicate of trials with group-reared embryos in 24-well plates as stated in the OECD TG 236 FET test guideline to underline the validity of your findings even with changing rearing conditions.

The overall language used in the article is professional and clearly understandable. If possible, it could be re-checked by a native speaker or professional editor for further improvements.

Reviewer 3 Report

The authors evaluated the embryotoxicity of nicotine and cytisine (a drug used in the pharmacological treatment against nicotine addiction) in zebrafish. They reported that nicotine is more embryotoxic than cytisine and a partial reduction in the embryotoxicity of nicotine during the co-exposure with cytisine.

I find the main aim and scope of the paper interesting. However, I am not very sure whether the design was the most appropriate and whether the results are fully conclusive due to the lack of dose-response in several of the effects reported by the authors. Therefore, to my understanding the manuscript needs to address certain issues before publication.

I do not understand the choice of 0.5-, 0.75- and 1.0-mM nicotine for experiments of co-exposure experiments with cytisine. These doses are approximately equal and significantly higher than the LC50 and therefore extremely toxic. What is the point of testing teratogenicity at a dose that causes 100% lethality?

According to Figure 1B there was approximately 70% of hatching after exposure to 1 mM nicotine. How does this square with the 100% mortality reported in Figure 3A?

Given the lack of dose response (Figure 2D), I think that the effect of cytisine on pigment depletion is at least questionable.

Figure 3A: There is a lack of dose-response in the percentage of malformations; which is even lower than the control and 0.5 mM. This does not seem to be explained by mortality. I think that Figure 3A would be improved with a statistical evaluation, which is currently lacking.

Again, there is a lack of dose response in the attenuation of mortality when nicotine and cytisine are co-administered. One can understand that 1 mM cytisine could reduce the mortality caused by 0.75 mM nicotine by 30%, but it is more difficult to understand that the same concentration of cytisine could reduce the mortality of 1 mM nicotine by 83%.

In Figure 3C, D and E the statistical comparison of the control with other experimental conditions is interesting and convenient. However, I missed the really relevant ones; which are the comparisons of nicotine versus nicotine + cytisine.

There is a lack of dose-response in Figure 3C because the heart rate of 0.75 mM nicotine + cytisine is lower than the heart rate of 1 mM nicotine + cytisine.

In the discussion, the authors explain the observed effects based on differences in the affinity of nicotine and cytisine for different nicotinic acetylcholine receptors. Are the different types of these receptors and their sensitivity to these agonists comparable between zebrafish and humans?

MINOR POINT

In Figure 1B, I missed the percentage (8%) of unhatched larvae in the 0.5 mM bar. Please, note that these percentages are shown in the 0.75 and 1 mM bars.

Round 2

Reviewer 2 Report

Dear authors, thank you for considering the comments and suggestions I have made during the review process. Other than the separate rearing of the embryos, where i still find no statement within the article about how you ensure that this does not affect your results, I see no futher issue with the manuscript. If you could include one/two sentences to this subject in the mansucript, that would greatly support the results and discussion.

Author Response

Comment: Dear authors, thank you for considering the comments and suggestions I have made during the review process. Other than the separate rearing of the embryos, where i still find no statement within the article about how you ensure that this does not affect your results, I see no futher issue with the manuscript. If you could include one/two sentences to this subject in the mansucript, that would greatly support the results and discussion.

Respons:

Dear Reviewer thank you very much for your thoughtful review and positive acceptance of the response. Your feedback is highly valuable and will help us improve the quality of our work. We also included the statement in the Method section

"Although zebrafish are are social animals, and their interactions with other fish can influence their behavior and physiological response we may suggest that separate rearing of zebrafish embryos did not  affect the results  of FET, as it was prevoiusly revealed [38].
To address these concerns and minimize bias, we implamented  following strategies: randomization and  appropriate control groups that are raised and treated identically to the experimental groups, except for exposure to the toxic agent, is crucial for establishing a baseline for comparison"

  1. Jarema, K.A.; Hunter, D.L.; Hill, B.N.; Olin, J.K.; Britton, K.N.; Waalkes, M.R.; Padilla, S. Developmental Neurotoxicity and Behavioral Screening in Larval Zebrafish with a Comparison to Other Published Results. Toxics. 2022 17;10(5):256.

Reviewer 3 Report

The authors have made a significant number of changes to the paper, addressing most of the concerns I raised in my first report. Other concerns have been addressed, in particular those related to the lack of a dose-response, and these have also been successfully addressed in the rebuttal letter. Overall, I have no further objections and recommend that the manuscript be published in its current form.

Author Response

Dear Reviewer thank you very much for your thoughtful review and positive acceptance of the response. Your feedback is highly valuable and will help us improve the quality of our work.